# Use of Transradial Access to Install Two Sequential Stents for Pseudoaneurysms along the Celiac Artery and Common Hepatic Artery Axes

**DOI:** 10.3390/diagnostics13203273

**Published:** 2023-10-21

**Authors:** Abheek Ghosh, Sean Lee, Christina Lim, Tanvir Agnihotri, Nabeel Akhter

**Affiliations:** 1Department of Diagnostic Radiology and Nuclear Medicine, University of Maryland Medical Center, Baltimore, MD 21202, USA; 2Department of Basic Biomedical Sciences, Touro College of Osteopathic Medicine, Middletown, NY 10027, USA; 3Department of Basic Biomedical Sciences, Creighton University, Omaha, NE 68178, USA; 4University of Pennsylvania, Philadelphia, PA 19104, USA; 5Department of Vascular and Interventional Radiology, Mercy Medical Center, Baltimore, MD 21202, USA

**Keywords:** transradial access, transfemoral access, stenting, visceral pseudoaneurysms

## Abstract

Transfemoral access is the most common method for stenting visceral aneurysms. Over the years, transradial access has gained tremendous traction in interventional procedures due to many reported benefits, including increased patient comfort, decreased procedural cost, and reduced rates of procedural complications, among others. Moreover, transradial access can serve as a valuable alternative when transfemoral access may be contraindicated. Here, we successfully utilized transradial access to sequentially place two stents for pseudoaneurysms in the celiac artery and common hepatic artery.

Transradial access (TRA) has gained significant popularity over the years in body interventional procedures. Many studies have reported the benefits of TRA over transfemoral access (TFA) [1]. These factors include increased patient comfort, reduced cost, faster postoperative turnaround, and a decreased rate of access site complications, among others [1]. Several disadvantages of TRA, when compared to TFA, include slightly higher radiation exposure and an increased rate of access failures [2,3]. Nonetheless, TRA is a valuable tool to have within the interventionalists’ arsenal in cases where TFA may be contraindicated (i.e., tortuous iliofemoral vessel anatomy or major coagulopathic considerations) [4,5].

Although quite rare in occurrence, visceral artery aneurysms and pseudoaneurysms, which typically affect the celiac, superior, or inferior mesenteric arteries, can present a significant risk of life-threatening hemorrhage upon rupture [6,7]. As a result, treatment modalities like stenting, for instance, are key in preventing patient complications. Traditionally, endovascular stenting is conducted via TFA, mostly due to overwhelming familiarity with this technique and hesitance to use TRA. Here, we successfully utilize TRA to create two sequential stents for pseudoaneurysms along the celiac artery (CA) and common hepatic artery (CHA) axes.

A 58-year-old male with a history of pancreatic cancer s/p recent distal pancreatectomy and splenectomy with Nanoknife ablation presented with hemodynamic instability and decreasing hemoglobin (11 g/dL to 9.6 g/dL). Contrast CT findings showed evidence of an intraperitoneal bleed most likely attributable to pseudoaneurysms along the CA and CHA axes. Due to the patient’s coagulopathic concerns and the operator’s preference, the decision was made to pursue stenting along these planes via TRA. A sonogram of the left radial artery (RA) was performed and revealed that the patient had a Barbeau Type B waveform. With a micropuncture needle, the left RA was accessed under ultrasound guidance. A 5 French (Fr) slender sheath (10 cm) was placed over a 0.018 wire; heparin was given via the sheath. A 5 Fr pigtail (100 cm) catheter was placed into the abdominal aorta over a Bentson wire (145 cm) and an abdominal aortogram was then performed. A catheter was exchanged over a wire for a 5 Fr Ultimate catheter (120 cm), and the superior mesenteric artery was then selected. A superior mesenteric arteriogram was performed. The celiac axis was selected, and a celiac axis arteriogram was performed (Figure 1). A Truselect microcatheter (155 cm) with the help of a fathom wire (180 cm) was placed into the common and right hepatic artery (HA); common and right hepatic arteriograms were then performed. A microcatheter was placed deep into the right HA branch (Figure 2). A V18 exchange-length wire (300 cm) was placed through the microcatheter into the right HA branch (Figure 3). A microcatheter, parent catheter, and a 5 Fr sheath were removed, and a 6 Fr radial sheath (10 cm) was then placed over the wire. Two 6 mm x 5 cm covered self-expanding Viabahn stents were then sequentially placed. The first stent was placed covering the pseudoaneurysm in the CHA and a second stent was placed extending from the first stent into the celiac axis (Figure 4 and Figure 5). The Ultimate catheter (120 cm) was then placed into the celiac axis and a post-stenting celiac axis arteriogram was later performed. The catheter, wire, and sheath were removed, and a pressure bandage was inflated for patent hemostasis of the left RA. Sterile dressings were applied. The patient tolerated the procedure well without evidence of immediate complications. Post-stenting celiac axis arteriogram showed a patent celiac axis, patent left gastric artery, patent stents in the celiac axis, and CHA without visualization of an aneurysm, pseudoaneurysm, or any extravasation with patent hepatic artery, right and left hepatic arteries, or branches. Later, an abdominal aortogram and mesenteric angiogram also showed no evidence of active arterial extravasation.

Stenting is one of the main techniques used to treat visceral aneurysms and pseudo-aneurysms, which are associated with a significant risk of life-threatening hemorrhage upon rupture [6,7]. Most cases of stenting are traditionally performed via TFA; this is most likely due to the overwhelming familiarity with TFA among operators. However, in cases where TFA is not entirely favorable (i.e., coagulopathy or iliofemoral tortuosity), TRA may be a worthy alternative [6,7]. Apart from the aforementioned benefits (i.e., increased patient comfort or faster recovery times), TRA is also associated with a decreased rate of procedural complications when compared to TFA [1]. This is mostly attributable to the femoral artery being three times larger in diameter than the radial artery, boding a higher risk of complications such as bleeding, pseudoaneurysm, or AV fistula formation. While TRA has its benefits, it also comes with disadvantages, with the most notable one being increased radiation exposure [2,3]. However, studies that show this trend predominantly refer to cardiovascular interventions, which entail the TRA catheter traversing against the direction of flow of the ascending aorta, inevitably adding resistance. On the other hand, in body interventional procedures, the TRA catheter goes in the same directional flow as the descending aorta, presenting less counter-resistance than seen in cardiac interventions. This rationale is upheld by several body investigations which actually report lower radiation exposure in TRA than in TFA [1]. It is important to highlight that TRA may not be accessible in all patients (i.e., those with a Barbeau Type D waveform) and so TFA is the preferable option for these individuals. All things considered, our successful stenting within this case study helps show that TRA may serve as a valuable arterial access modality for treating aneurysms or pseudoaneurysms, among other clinical conditions.

## Figures and Tables

**Figure 1 diagnostics-13-03273-f001:**
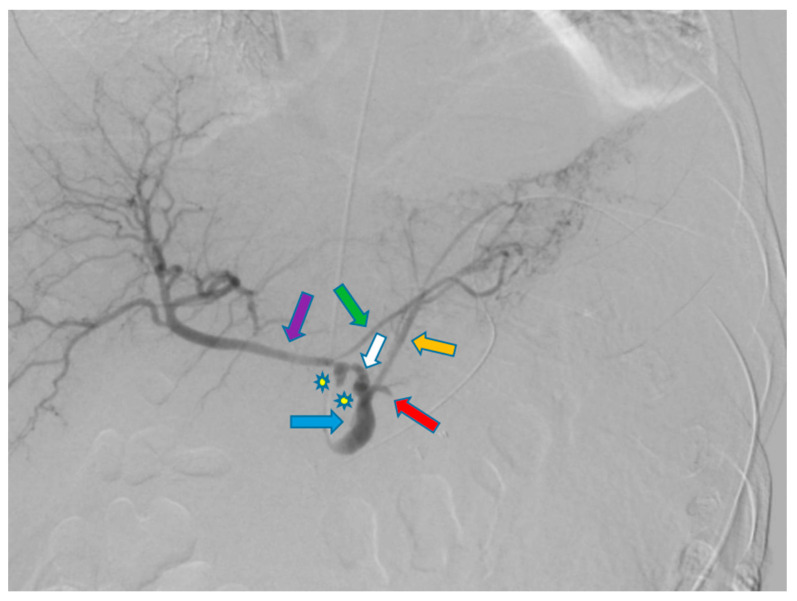
Celiac arteriogram shows the presence of two pseudoaneurysms (stars) along the celiac artery (blue arrow) and common hepatic artery (white arrow). Other notable vessels include the left gastric artery (orange arrow), proper hepatic artery (purple arrow), right gastric artery (green arrow) and splenic artery (red arrow) previously ligated at time of splenectomy.

**Figure 2 diagnostics-13-03273-f002:**
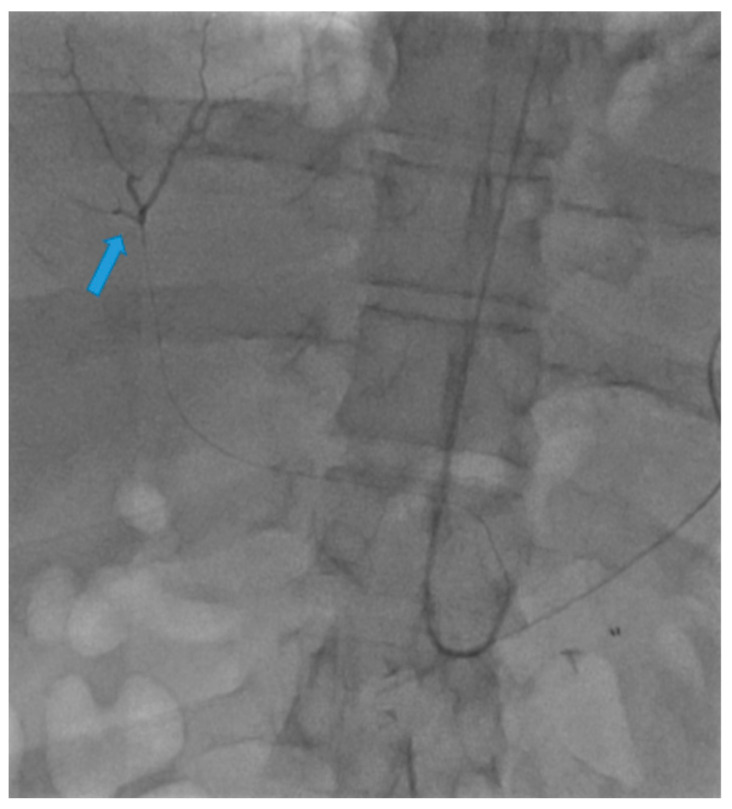
A Truselect microcatheter with the help of a fathom wire was placed into the common and right hepatic artery (HA); common and right hepatic arteriograms were then performed. A microcatheter was placed deep into the right HA branch (blue arrow).

**Figure 3 diagnostics-13-03273-f003:**
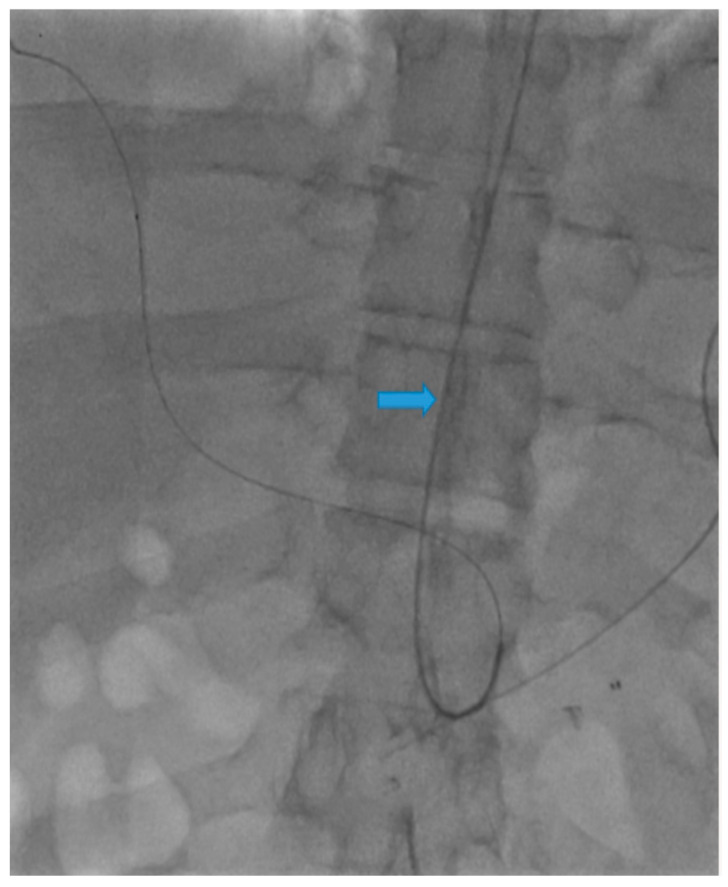
A V18 exchange-length wire (blue arrow) was placed through the microcatheter into the right HA branch.

**Figure 4 diagnostics-13-03273-f004:**
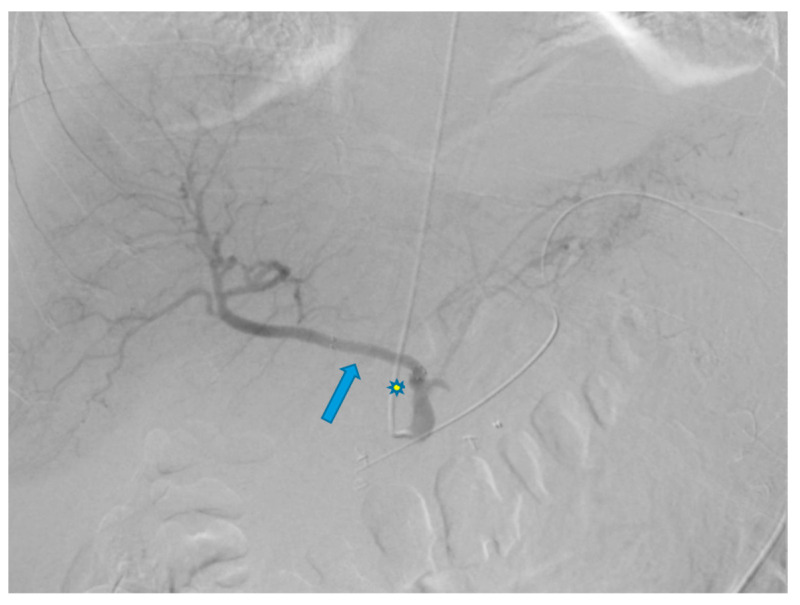
Arteriogram of the common hepatic artery after placement of the first stent (blue arrow) shows resolution of the pseudoaneurysm in the common hepatic artery but persistence of the pseudoaneurysm (star) in the celiac axis.

**Figure 5 diagnostics-13-03273-f005:**
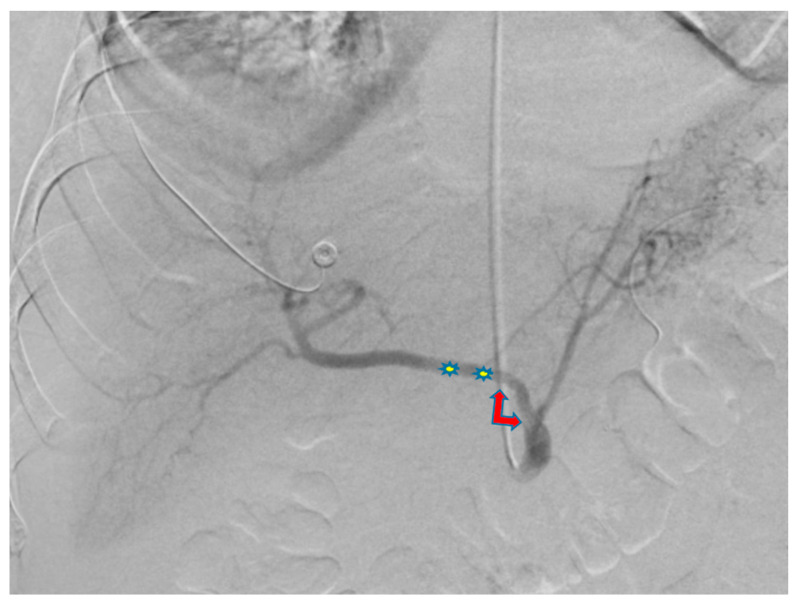
Celiac arteriogram after placement of both stents (stars) shows resolution of the pseudoaneurysms (double-headed arrow). This arteriogram shows a patent celiac axis, patent left gastric artery, patent stents in the celiac axis, and CHA without visualization of an aneurysm, pseudoaneurysm, or any extravasation with patent hepatic artery, right and left hepatic arteries, or branches.

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
