# Peer review of "Use of Transradial Access to Install Two Sequential Stents for Pseudoaneurysms along the Celiac Artery and Common Hepatic Artery Axes"

_diagnostics, 2023, doi:10.3390/diagnostics13203273_

Round 1

Reviewer 1 Report

Dear editor,

I read with interest the article written by Abheek Ghosh et al., in which they presented the endovascular resolution of two pseudoaneurysms along the celiac artery and common hepatic artery in a 58-year-old male patient with a history of pancreatic cancer for which he recently underwent distal pancreatectomy and splenectomy with Nanoknife ablation.

The case is very clearly presented, with numerous qualitative figures, making it easy to read and follow.

The endovascular resolution of pseudoaneurysms is very well performed, being of interest for current practice.

In my opinion, this article meets all the necessary criteria to be published as "interesting images" in its current form.

Author Response

Dear MDPI Reviewer,

We hope this letter finds you well. We would like to express our sincere gratitude to you and the team of reviewers for dedicating your time and expertise to thoroughly evaluate our manuscript titled "Use of transradial access to install two sequential stents for pseudoaneurysms along the celiac artery and common hepatic artery axes." We appreciate your insightful comments. 

Reviewer 2 Report

The authors presented a case of treatment of pseudoaneurysms of the common hepatic artery using covered stents via radial access.

In our center, lesions in the visceral arteries are treated, mainly the celiac trunk and its branches - but mainly the brachial approach is performed, less often radial access due to equipment incompatibility, e.g. when using a 110 cm long sheath, there is a problem with stents where the working part is up to 120 cm, in tall patients, distal location of the lesion, etc.

Comments/questions

- please provide the exact dimensions of the equipment used in this case - length, type of catheters, sheath;

- why were such long stents used - the lesion to cover is about 3-4 cm - why were 2 stents of 5 cm each used - wouldn't 1 be enough?

- is the image inverted in Figures 2 and 3 - does the explanation under the figure include information that a microcatheter and then a guidewire were placed in the right hepatic artery (and the image suggests a location on the left)?

- what are the alternative solutions - e.g. a covered stent cannot be provided and what then - e.g. stent + embolization coils?

Author Response

Dear MDPI Reviewer,

We hope this letter finds you well. We would like to express our sincere gratitude to you and the team of reviewers for dedicating your time and expertise to thoroughly evaluate our manuscript titled "Use of transradial access to install two sequential stents for pseudoaneurysms along the celiac artery and common hepatic artery axes." We appreciate your insightful comments and recommendations. Attached are our responses to each inquiry. 
